# Using Wearable Inertial Sensors to Monitor Effectiveness of Different Types of Customized Orthoses during CrossFit^®^ Training

**DOI:** 10.3390/s23031636

**Published:** 2023-02-02

**Authors:** Lorenzo Brognara, Antonio Mazzotti, Federica Rossi, Francesca Lamia, Elena Artioli, Cesare Faldini, Francesco Traina

**Affiliations:** 1Department of Biomedical and Neuromotor Sciences, University of Bologna, 40123 Bologna, Italy; 21st Orthopaedic and Traumatologic Clinic, IRCCS Istituto Ortopedico Rizzoli, 40136 Bologna, Italy; 3Data Analyst, Stat.Sci, University of Bologna, 40136 Bologna, Italy

**Keywords:** inertial sensor, postural balance, risk factors, stability, instability, foot orthosis

## Abstract

Background: Dynamic balance plays a key role in high-impact sports, such as CrossFit, where athletes are required to maintain balance in various weightlifting exercises. The loss of balance in these sport-specific movements may not only affect athlete performance, but also increase the risk of injuries. Objectives: The aim of the study is to achieve greater insight into the balance and athlete position during the CrossFit training by means of inertial sensors, with a particular focus on the role of different custom foot orthoses (CFOs) in order to detect correlations with the role of the cavus foot. Methods: A total of 42 CrossFit^®^ athletes, aged 25 to 42 years, were enrolled in this study. One-way ANOVA tests with post-hoc analysis of variance were used to compare foot posture groups and effects of different types of customized foot orthoses. Results: When comparing the effects of CFOs with the respective balance basal level during the pistol squat exercise, we observed a significant (*p* = 0.0001) decrease in the sway area, antero-posterior displacement (APD) and medio-lateral displacement (MLD) compared to the basal using both types of CFOs. Conclusion: No significant positive effects of CFOs were observed in some static tests. On the contrary, positive effects of CFOs and, in particular, postural insoles, are relevant to dynamic balance.

## 1. Introduction

During the last few years, CrossFit has gained popularity in recreational and competitive forms, with a large percentage of musculoskeletal injuries; however, the biomechanical analysis of athlete sport-specific movement has not yet been studied [1]. Given the growing popularity of CrossFit^®^ (globally there are over 15,000 affiliated CrossFit gyms and over 4 million participants estimated), proper education is necessary for athletes in order to avoid possible risks arising from bad postural balance during training. Workouts are made up of different exercises, including metabolic conditioning (running or skip rope), gymnastics (pull-ups, burpees, etc.), Olympic weightlifting (snatch and clean and jerk) and powerlifting movements (squat, bench press and deadlift).

Recent technological developments ensure to health figures and sports scientists new tools to validate rehabilitation treatment in a more objective way. In the last few years, there is a growing interest for developing technologies and methods for enabling human motion analysis that allows an understanding of the locomotor demands of various sport-specific movements. Sport scientists are focusing more than ever on products that deliver cheap, faster and more efficient clinical and biomechanical evaluation. Postural balance is evaluated by several methods and tools. In terms of computerized testing force, platforms are the most used methods for quantitative analysis of balance; however, they are expensive instruments and with several methodological limitations, which restrict their widespread sport and clinical application [2].

Inertial sensors provide a cheap and accessible means to efficiently collect and process large amounts of athlete balance data in an unconstrained environment (outside of the laboratory environment) on the field and during dynamic tasks (such as sport-specific movements). The role of wearable sensors in postural analysis is becoming more important in sports and clinical applications [3,4,5,6,7,8]. Thanks to an inertial sensor with a dedicated protocol, it was possible to detect sport-specific movements, allowing coaches and medical staff to understand the physical demands on sport athletes. The ability of inertial sensors to capture sport-specific movements provides further detail on athlete demands and performance. Most commercially available wearable sensor units contain microsensors that include the use of gyroscopes, accelerometers and magnetometers. Some authors (such as Paillard, Zemková and Hamar) found that short and intensive general exercise increases postural sway when the energy expenditure exceeds the lactate accumulation threshold or the athlete hyperventilates [9,10]. In this research, validation of the hypothesis has been developed by comparing CrossFit sport-specific movements performed by the athletes with different types of custom foot orthoses (CFO). The CrossFit training shows peculiar characteristics, including the motivational scalability of exercises, the self-challenge among the pairs and personal records and the adoption of a healthy lifestyle. Physical exercise programs come from gymnastics, weightlifting and calisthenics; they are characterized by high volume and intensity, with short or no rest periods. Functional exercises are characterized by constant variation: from an aerobic exercise (i.e., running) and calisthenics to power/Olympic weightlifting [11,12]. Foot orthoses have been broadly used by clinicians to treat foot impairments and sports-related lower limb injuries; in fact, while it is not uncommon for athletes to use foot orthoses to relieve foot pain, little is known about their effects on sports performance and balance. However, the relationship among dynamic stability and the use of foot orthoses on sport movements is not yet established [13].

Participants were randomized to receive a biomechanical insole or a postural insole and following a 4-week washout period, the participants crossed over to the other treatment. Even though many studies have investigated the effect of insoles on balance, with benefits on postural balance and control, as far as we know, no studies have yet investigated the effect of insoles on balance using wearable inertial sensors during sport-specific movements [14,15].

## 2. Materials and Methods

### 2.1. Participants

Forty-two CrossFit athletes was recruited between October 2021 and July 2022. In accordance with the requirements established by the declaration of Helsinki, written consent was obtained from each person, after having been informed in a clear and simple way about the purpose of the study and the procedures involved. The study protocol was approved by the Human Research Ethics Committee (reference: n° 659/2021/Sper/IOR). Inclusion criteria were subjects of both sexes who fulfilled the following criteria: (1) age of 18 years or older; (2) cavus feet with an arch index < 21% and Clarke’s angle > 45° bilaterally (thresholds were measured with clinical functional evaluation and baropodometric analysis); (3) CrossFit athletes for at least 3 years. The criteria for qualifying for the study included being aged from 18 to 43 years, in good health with a valid and up-to-date medical certificate, at least 3 years of regular CrossFit training experience and a minimum of 4 workout sessions (CrossFit) per week. We included both males and females. Exclusion criteria were: (1) history or presence of musculoskeletal conditions that may alter balance and/or gait; (2) lower limb injuries in the previous 6 months; (3) history of orthopedic surgery; (4) recent (<3 months) hospitalization due to the diagnosis of COVID-19. All the athletes provided demographic information, medical history and previous treatments for foot diseases. The characteristics of the participants are showed in Table 1. All the participants had self-reported no lower extremity injuries that required medical consultation and/or sports participation disruption for longer than two weeks in the last three months prior to study participation.

### 2.2. Inertial Sensor Measurement System (IMUs)

Data are collected with an inertial sensor measurement system (IMUs) consisting of one sensor unit positioned at L5 level (Figure 1). We used a reliable and validated IMUs wearable posturographic sensor system (mSway, mHealth Technologies, BO, Bologna, Italy). IMUs allow to obtain several postural parameters and estimate, with great accuracy, the balance parameters (Table 2); as well as position, acceleration and speed produced by the movement IMU, which is made up of a transducer, known as a micro-electro-mechanical system (MEMS) that detects movement and transforms the mechanical signal into an electrical one using an algorithm. The transducer was composed of a tri-axial accelerometer, tri-axial gyroscope and magnetometer, which can provide quantitative data about patient performance that can be useful to improve clinical interpretation and to evaluate treatments in a more objective way. A wearable sensor is able to stop automatically after the exercise and the raw data of the inertial sensors are led back on the CrossFit exercises.

### 2.3. Examination Procedures

The study selects athletes with a pes cavus (high arch), based on previous studies that suggest a functional correlation between a higher arch height and the probability of injuries [16,17,18,19,20]. Cavanagh and Rodgers showed a method of measuring footprints for classifying foot types [21]. They defined the arch index as a ratio of the area of the middle third of the footprint to the entire footprint area. They suggested criteria for the classification of footprints with a high arch (arch index < 0.21), normal arch (0.21 < arch index < 0.26) and flat arch (arch index > 0.26). Athletes, after receiving a physical, podiatric and orthopedic examination, were randomly given a biomechanics CFO and postural CFO (Figure 2) using a random number table. Biomechanical CFO insoles are to correct deformities and improve foot function by supporting the medial longitudinal arch or throughth wedges [22]. Postural (proprioceptive) insoles, with no arch support or wedges, simulate correction reflexes throught activation of ascending proprioceptive chains and muscle proprioception that are collected primarily by cutaneous foot mechano-receptors [23]. The experimental protocol consisted of a static and dynamic (sport-specific movements) analysis under two experimental conditions (with and without the orthoses) using wearable inertial sensors mSway (mHealth Technologies, Bologna, Italy) previously validated [24].

Postural/balance assessment was analyzed by measuring the motion of sway of the body, the antero-posterior displacement (APD) and medio-lateral displacement (MLD), as shown in Table 2. The static tests were performed standing with feet close together or tandem and each proof was performed with or without visual perturbation (eyes open or closed) and somatosensory perception disturbances (foam surface). Static and dynamic tests (over-head squats and pistol squats) were performed before and after the use of different insoles. Following a 4-week washout period, participants crossed over to the other treatment, as shown in Table 3 and Figure 3.

### 2.4. Statistical Analysis

Data were presented as mean and standard deviation (SD); the quantitative variables were subjected to a descriptive analysis using central tendency and dispersion measures (mean and standard deviation). Interquartile range (IQR), coefficient of variation, standard deviation index (σ) and the ANOVA test are used to estimate the distribution of quantitative variables. The differences between the means of the two CFOs are also analyzed using graphic distributions (boxplot). Tests are considered significative if *p* < 0.05.

## 3. Results

The analysis of the dynamic tests shows a more significant effect of CFOs (pr < 0.0001). In the pistol squat (sport-specific movement), both orthotic therapies were significantly lower than the average of the oscillations from the first use with a stable improvement. ANOVA was used to determine statistical differences in the balance oscillations between the different tests and CFO (Table 4 and Table 5). The dispersion indexes measured over time show that the proprioceptive insole is superior to the biomechanical insole. The PS and OHS tests prove a statistically significant benefit; in addition, the use of the insole determines an immediate improvement that persists in the medium term: T0 vs. T1 and T2 (Figure 4).

As illustrated on the study flow diagram in Table 3, T0 represents tests performed without CFOs; and T1, T2, T3 and T4, tests performed with CFOs. It can be noticed in the boxplots on Figure 4 (that is a special type of diagram that shows the quartiles in a box) that there is a reduction in the mean and variability (IQ range) of the oscillations (SA, DML, APD), especially in the pistol-squat (PS) movement from T0 (test performed without CFOs) to T1, T2 and T3 (test performed with CFOs). In fact, in each plot it is possible to observe the influence of the biomechanical CFO (T1 and T2) and postural CFO (T3, T4) on different postural parameters, comparing CrossFit sport-specific movements (dynamic tests: OHS and PS) performed by the athletes with an inertial sensor worn on the body.

## 4. Discussion

The dynamics of the human body during sport-specific movements has generated considerable recent research interest among scientists dedicated to reducing the number of injuries and for performance improvement. The main objectives of this paper were to provide new insights about the postural sport assessment through a wearable inertial sensor with a particular focus on the role of different custom foot orthoses (CFOs) on balance performance. CrossFit sport-specific movements, such as the pistol squat (a highly challenging body-weight movement that tests the balance, coordination, flexibility and strength of athletes) and the overhead squat (a difficult lift exercise that requires upper-body strength and balance), were analyzed using wearable sensors that could constitute an interesting set up for in-field CrossFit movement analysis, being able to detect the role of the insole more accurately.

Thanks to an inertial sensor, it is possible to better understand the physiological and balance changes induced by exercises to provide the basis for designing better therapies, such as custom foot orthoses or balance training programs, tailored to individual athletes with the goal of reducing injuries. The application of inertial sensor technology in the assessment of balance is a new and evolving domain useful to establish a new field of dynamic balance assessments in sports medicine. Regarding the inertial sensor set up, one of the main advantages was that it was not necessary to perform a calibration before conducting the data collection during the training. In fact, thanks to the algorithm, during the postprocessing, the raw data of the inertial sensors were reconstructed based on the design of the different CrossFit exercises [25]. The real clinical impact of custom foot orthoses therapy in sports remains controversial and is not yet fully understood. While the link appears to sometimes lead to positive outcomes, as in the case of patellofemoral pain or plantar fascitis, it is certain that studies with a more appropriate comparison in the relationship of the effectiveness on sport-specific movements are needed [26,27]. Not only training preparation, but also the balance is connected with injury prevention: postural control can be described as either dynamic or static; static postural control is attempting to maintain a base of support while minimizing the movement of the center of mass, while dynamic postural control involves the achievement of a functional task with sport-specific movements without compromising an established base of support [28]. Therefore, it is essential to optimize the training process in terms of postural stability to be as efficient as possible and to avoid possible injury and chronic pain syndromes [29,30]. It is essential to understand the risk factors such as an abnormal foot alignment for athletes who perform high-intensity power training before devising proper custom foot orthoses and further research is needed to improve the clinical outcomes of athletes who suffer injury due to an impaired balance [31,32,33,34]. Statistical results obtained by the methodology developed encourage the further development of the proposed approach.

### Limitations and Future Direction

Studies with larger samples, which include in their methods different training sessions with different exercise intensities and duration, are still needed to better investigate the benefits of foot orthoses on postural stability and draw more reliable conclusions.

This study is not without limitations: one limitation of the present study concerns the evaluation of effects of foot orthoses on postural balance; the long-term effects were evaluated at 3 weeks. For these reasons, further studies should evaluate the effects at one year in order to better understand the balance improvement over time and the long-term clinical impact of this rehabilitative approach.

## 5. Conclusions

Balance is one of the limiting factors of performance in many sports and there is a clear need of studies that analyze the effects of sport-specific exercises on postural balance control in CrossFit athletes. To our knowledge, the present research may be considered as the first study applying wearable inertial sensor in order to investigate the CrossFit movement during training and furthermore, the possible correlations with the use of CFOs, being important therapy for injury prevention and balance improvement. Both the insoles (biomechanical and postural) produced a significant decrease of sway area, APD and MLD during the pistol-squat exercise and postural insole, even on the overhead squat. These changes may lead to an overall improvement in balance pattern and stability, suggesting that the use of these types of insoles may be a useful treatment strategy for improving dynamic balance in CrossFit athletes. Movement of the foot and ankle influences balance and it is important to understand the complex foot rotatory forces that occur during sports exercises in order to detect impaired lower limb biomechanics among people who participate in power/Olympic weightlifting sports such as CrossFit. CFOs should help to control foot and ankle joint moments; in addition, we demonstrated that customized foot orthoses may be a useful treatment strategy for improving foot alignment, controlling subtalar joint movement and excessive supination and reducing postural sway. Healthcare professionals and sport scientists should consider CFOs a key part of a treatment plan for sports injury prevention and the wearable sensor as a beneficial tool useful to assess the balance and performance improvements during the sport-specific movements.

Based on the results, one could possibly assume that foot orthotics represent one key factor that may help to reduce the risk of lower limb injury due to a balance improvement during the sport-specific exercises; however, the degree of benefit and which disease states respond to this treatment require further investigation.

## Figures and Tables

**Figure 1 sensors-23-01636-f001:**
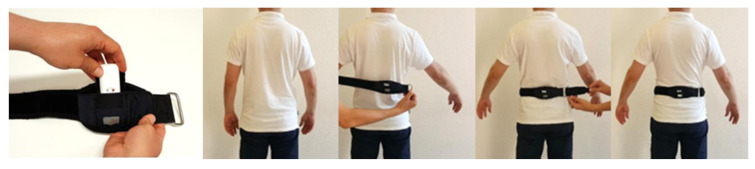
Inertial measurement unit (IMU) and applications.

**Figure 2 sensors-23-01636-f002:**
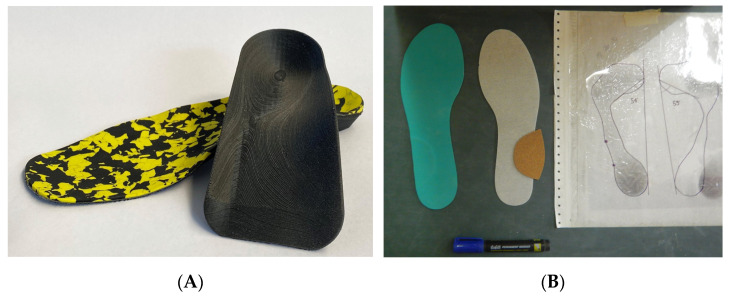
All the participants had negative plaster casts taken in subtalar joint neutral position with the participant in supine position. Biomechanical CFO (**A**) were made of a 2 mm thick carbon shell with a straight extrinsic ethylene-vinyl-acetate (EVA) rearfoot post. Postural CFO (**B**) were made with a half-moon anti-varus elements (cork) having a 1.5 mm thickness.

**Figure 3 sensors-23-01636-f003:**
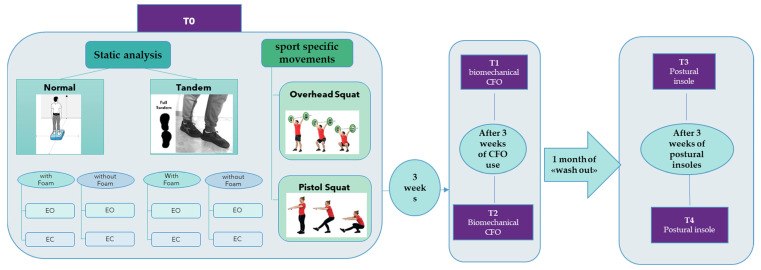
Graphical representation of the testing procedure. EO: eyes open; EC: eyes closed; CFO: custom foot orthoses; OHS: overhead squat; PS: pistol squat.

**Figure 4 sensors-23-01636-f004:**
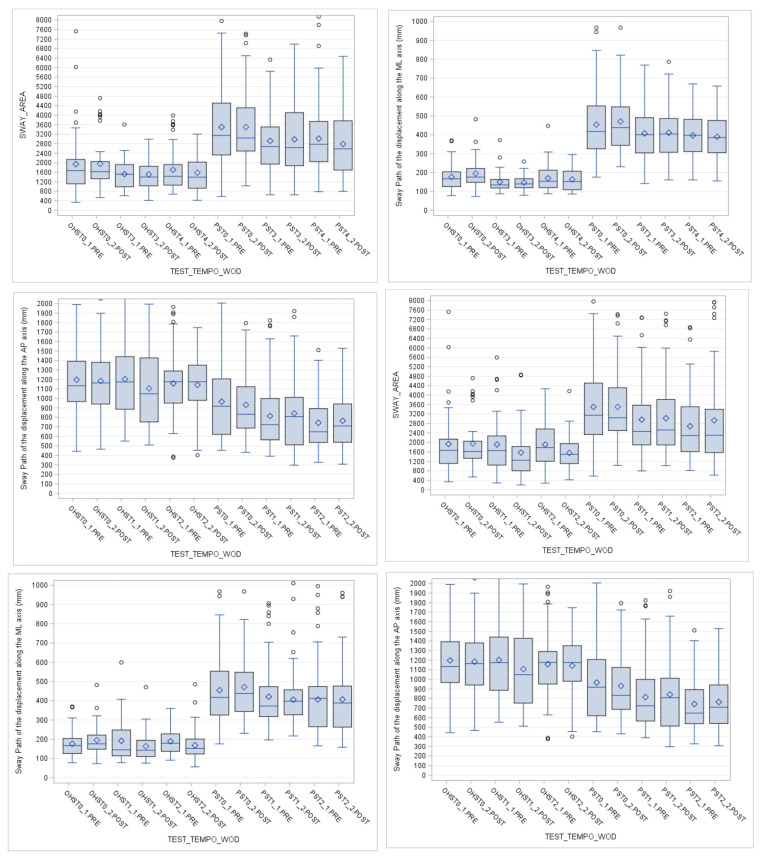
In the following figure is shown an example of how both CFOs affected the sway area indexes in the dynamic movement PS, reducing balance oscillations. Boxplots represent principal dispersion indexes (IQ range, mean and median) values before and after the use of CFOs comparing the moment without CFO (T0) with the first use (T1 or T3) and medium term (T2 or T4) measurements. It can be noticed that there is a reduction in the mean and variability (IQ range) of the oscillations (SA, DML, APD), especially in PS movement. The reduction in OHS is less significant.

**Table 1 sensors-23-01636-t001:** Mean ± SD of anthropometric and demographic data features of CrossFit Athletes.

Variables	Athletes (n = 42)	Male (n = 18)	Female (n = 24)
Age: mean (minimum-maximum)	32.5 (25–42)	31.3 (25–37)	34.1 (26–42)
Height (cm): mean (minimum-maximum)	174.9 (161–189)	181.3 (175–189)	166.5 (161–175)
Body mass (kg): mean (minimum-maximum)	71.7 (51–86)	80.75 (74–86)	59.6 (51–69)
Shoe size: mean (minimum-maximum)	42 (37–45)	43 (41–45)	39 (37–42)

**Table 2 sensors-23-01636-t002:** Definition of the parameters used in the assessment of balance.

Parameters	Definition
Sway area [mm^2^/s]	Displacements area of the center of mass (CoM) in the unit of time.
Antero-posterior displacement (APD) [mm]	Anteo-posterial displacement area of the center of mass (CoM)
Medio-lateral displacement (MLD) [mm]	Medio-lateral displacement area of the center of mass (CoM)

**Table 3 sensors-23-01636-t003:** Description and flow diagram of testing procedure.

**T0 static—Without insole**				**T1 static—With biomechanical insoles**				**T3 static** **—With postural insoles**			
#1	** *EO* **	30 sec	***NO*** foam	#1	** *EO* **	30 sec	***NO*** foam	#1	** *EO* **	30 sec	***NO*** foam
#2	** *EC* **	30 sec	***NO*** foam	#2	** *EC* **	30 sec	***NO*** foam	#2	** *EC* **	30 sec	***NO*** foam
#3	** *EO, tandem* **	30 sec	***NO*** foam	#3	** *EO, tandem* **	30 sec	***NO*** foam	#3	** *EO, tandem* **	30 sec	***NO*** foam
#4	** *EC, tandem* **	30 sec	***NO*** foam	#4	** *EC, tandem* **	30 sec	***NO*** foam	#4	** *EC, tandem* **	30 sec	***NO*** foam
#5	** *EO* **	30 sec	**with** foam	#5	** *EO* **	30 sec	**with** foam	#5	** *EO* **	30 sec	**with** foam
#6	** *EC* **	30 sec	**with** foam	#6	** *EC* **	30 sec	**with** foam	#6	** *EC* **	30 sec	**with** foam
#7	** *EO, tandem* **	30 sec	**with** foam	#7	** *EO, tandem* **	30 sec	**with** foam	#7	** *EO, tandem* **	30 sec	**with** foam
#8	** *EC, tandem* **	30 sec	**with** foam	#8	** *EC, tandem* **	30 sec	**with** foam	#8	** *EC, tandem* **	30 sec	**with** foam
**T0 sport-specific movements** **—Without insoles**				**T2 sport-specific movements** **—With biomechanical insoles**				**T4 sport-specific movements** **—With postural insoles**			
#1	** *OHS* **	empty barbell		#1	** *OHS* **	empty barbell		#1	** *OHS* **	empty barbell	
#2	** *OHS* **	empty barbell		#2	** *OHS* **	empty barbell		#2	** *OHS* **	empty barbell	
#3	** *OHS* **	empty barbell		#3	** *OHS* **	empty barbell		#3	** *OHS* **	empty barbell	
#4	** *PS* **	right leg		#4	** *PS* **	right leg		#4	** *PS* **	right leg	
#5	** *PS* **	left leg		#5	** *PS* **	left leg		#5	** *PS* **	left leg	
#6	** *PS* **	right leg		#6	** *PS* **	right leg		#6	** *PS* **	right leg	
#7	** *PS* **	left leg		#7	** *PS* **	left leg		#7	** *PS* **	left leg	
#8	** *PS* **	right leg		#8	** *PS* **	right leg		#8	** *PS* **	right leg	
#9	** *PS* **	left leg		#9	** *PS* **	left leg		#9	** *PS* **	left leg	

**Table 4 sensors-23-01636-t004:** Correlations between postural parameters and effects of orthoses (biomechanics CFO and postural CFO) on different tests (static and dynamic). NS = non-significant correlations. DEV.STD. = standard deviation.

		Biomechanical CFO	Postural CFO
Static test tandem	FOAM EC	sway was significantly decreased at T1 and T2	NS
	FOAM EO	NS	sway was significantly decreased at T3 and T4
	NO FOAM EC	sway was significantly decreased at T1 and T2	sway was significantly decreased at T3 and T4
	NO FOAM EO	NS	NS
Static test NO TANDEM	FOAM EC	sway was significantly decreased at T1 and T2	NS
	FOAM EO	sway was significantly decreased at T1 and T2	NS
	NO FOAM EO	NS	NS
	NO FOAM EC		
Dynamic test	PISTOL SQUAT	sway area, ADP and MLD were significantly decreased at T1 and T2	sway area, ADP and MLD were significantly decreased at T3 and T4
	OVERHEAD SQUAT	NS	sway area, ADP and MLD were significantly decreased at T3 and T4

**Table 5 sensors-23-01636-t005:** Mean and standard deviation of sway data during Pistol squat.

		Sway During Pistol Squat	Sway During Pistol Squat
		MEAN	DEV.STD.
**without CFOs**	**T0**	34.99	160.60
**Biomec. CFO**	**T1**	29.58	150.84
**Biomec. CFO**	**T2**	26.80	134.90
**Postural CFO**	**T3**	29.19	127.56
**Postural CFO**	**T4**	30.14	146.07

## Data Availability

The data presented in this study are available on request from the corresponding author.

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
