# Peer review of "Using Wearable Inertial Sensors to Monitor Effectiveness of Different Types of Customized Orthoses during CrossFit^®^ Training"

_sensors, 2023, doi:10.3390/s23031636_

Round 1
Reviewer 1 Report
The paper meets the journal objective. Having as purpose: study is to achieve greater insight into the balance and athlete position during the CrossFit training by means of inertial sensors, with a particular focus on the role of different custom foot orthoses (CFOs) in order to detect correlations with the role of the cavus foot.”
1) There are differences between the objective stated in the "Abstract" section and the "Discussion" section. The paper authors have to specify which is the correct one, in order to carry out a research effective evaluation.
2) There are other instruments that evaluate balance and position, it would be useful to mention them in the introductory and discussion section, as a way to theoretically base the choice of the inertial sensor, assessing advantages and limitations.
3) Support the idea of lines 241-243 with a cite from a directly related relevant research.
Author Response
- The paper meets the journal objective. Having as purpose: study is to achieve greater insight into the balance and athlete position during the CrossFit training by means of inertial sensors, with a particular focus on the role of different custom foot orthoses (CFOs) in order to detect correlations with the role of the cavus foot.”
The authors would like to thank the reviewer for recognizing the importance of the topic of our research and for any kind consideration
- There are differences between the objective stated in the "Abstract" section and the "Discussion" section. The paper authors have to specify which is the correct one, in order to carry out a research effective evaluation.
We are aware and agree with the reviewer. We clarified as suggested
L. 234.237
The main objectives of this paper were to provide new insights about the postural sport assessment through a wearable inertial sensor with a particular focus on the role of different custom foot orthoses (CFOs) on balance performance
- There are other instruments that evaluate balance and position, it would be useful to mention them in the introductory and discussion section, as a way to theoretically base the choice of the inertial sensor, assessing advantages and limitations.
According to Reviewer’s suggestion, we have improved the introduction and discussion section, we thank the reviewer for the useful suggestions.
Introduction
L.45-53
Postural balance is evaluated by several methods and tools. In terms of computerized testing force platforms are the most used methods for quantitative analysis of balance, but they are expensive instruments and with several methodological limitations which restrict its widespread sport and clinical application [2].
Inertial sensors, provide a cheap and accessible means to efficiently collect and process large amounts of athlete balance data in a an unconstrained environment (outside of the laboratory environment), on the field and during dynamic task (such as sport-specifing movements).
Discussion
L. 246-252
The application of inertial sensor technology in the assessment of balance is a new and evolving domain useful to establish a new field of dynamic balance assessments in sports medicine. Regarding the Inertial sensor set-up, one of the main advantages was that it was not necessary to perform a calibration before doing the data collection during the training. In fact, thanks to the algorithm, during the postprocessing, the raw data of the inertial sensors were reconstructed based on the design of the CrossFit different exercises [25].
- Support the idea of lines 241-243 with a cite from a directly related relevant research.
We did it, as suggested:
L.261-267
Therefore, it is essential to optimize the training process in terms of postural stability to be as efficient as potential and to avoid possible injury and chronic pain syndromes [29,30]. It is essential to understand the risk factors such as an abnormal foot alignment for athletes who perform a high-intensity power-training before devising a proper custom foot orthoses and further research is needed to improve the clinical outcomes of athletes who suffer of injury due to an impaired balance [31-34].
Reviewer 2 Report
The authors studied the effect of Inertial Sensor to monitor on different type of customized orthoses during different type of training. The research is interesting, however the manuscript need some modifications in order to be accepted for publication
The authors should provide a reference in relation of declaration of Helsinki
In page 5 there is a table” study protocol”. Please put the close caption of this table
What is the difference between both insoles (biomechanical and postural)?
In figure 4 is no easy to understand the result that the author want to show. I suggest to identify clearly every plot. Which one is using CFO and which one is using no CFO
The author should to rewrite the results.
Author Response
The authors studied the effect of Inertial Sensor to monitor on different type of customized orthoses during different type of training. The research is interesting, however the manuscript need some modifications in order to be accepted for publication
- The authors should provide a reference in relation of declaration of Helsinki
We thank the reviewer for the suggestions
L. 312-314
“Institutional Review Board Statement: The study was conducted in accordance with the Declaration of Helsinki, and approval of the Ethics Committee for Human Research at the University of Bologna (Reference: n◦ 659/2021/Sper/IOR; 22 July 2021).”
- In page 5 there is a table” study protocol”. Please put the close caption of this table
We are aware and agree with the reviewer. We changed the Figure 3. We believe now that, thanks to reviewer suggestions, the readability and the availability in the tables and figures information is useful to the reader
L. 178-179
Figure 3. (a) Flow-diagram and (b) Graphical representation of the testing procedure. EO: eyes open , EC: eyes close, CFO: Custom Foot Orthoses, OHS: Overhead Squat; PS: Pistol Squat.
- What is the difference between both insoles (biomechanical and postural)?
According to Reviewer’s suggestion we explained the differnces, we thank the reviewer for the useful suggestions
L.143-147
Biomechanical CFO insoles is to correct deformities and improve foot function by supporting the medial longitudinal arch or throughth wedges [22]. Postural (proprio-ceptive) insoles, with no arch support or wedges, simulate correction reflexes throught activation of ascending proprioceptive chains and muscle proprioception that are col-lected primarily by foot cutaneous mechano-receptors [23].
- In figure 4 is no easy to understand the result that the author want to show. I suggest to identify clearly every plot. Which one is using CFO and which one is using no CFO. The author should to rewrite the results.
We thank the reviewer for the suggestions; we rewrited the results clarifying the plots which refer to tests performed using CFO and without CFO.
L.222-23.
As illustrated on study flow-diagram in fig 3(a), T0 represent tests performed without CFOs and T1,T2,T3 and T4 tests performed with CFOs. Can be noticed, in Figure 4 on Box-plots (that is a special type of diagram that shows the quartiles in a box), a reduction in mean and variability (IQ range) of the oscillations (SA, DML, APD), especially in Pistol squat (PS) movement from T0 (test performed without CFOs) to T1, T2 and T3 (test performed with CFOs). In fact in each plot it is possible to observe the influence of biomechanical CFO (T1 and T2) and postural CFO (T3,T4) on different postural parameters, comparing CrossFit sport specific movements (dynamic tests: OHS and PS) performed by the athletes with an inertial sensor worn on the body.
We thank the reviewers for the useful comments and suggestions. The paper has been revised by taking all reviewers’ suggestions, we hope to have improved the manuscript.
Round 2
Reviewer 2 Report
no one